# Association between Physical Literacy and Self-Perceived Fitness Level in Children and Adolescents

**DOI:** 10.3390/biology10121358

**Published:** 2021-12-20

**Authors:** Raquel Pastor-Cisneros, Jorge Carlos-Vivas, Laura Muñoz-Bermejo, Jose Carmelo Adsuar-Sala, Eugenio Merellano-Navarro, María Mendoza-Muñoz

**Affiliations:** 1Promoting a Healthy Society Research Group (PHeSO), Faculty of Sport Sciences, University of Extremadura, 10003 Caceres, Spain; raquelpc@unex.es (R.P.-C.); jadssal@unex.es (J.C.A.-S.); mamendozam@unex.es (M.M.-M.); 2Social Impact and Innovation in Health (InHEALTH), University of Extremadura, 10003 Caceres, Spain; lauramunoz@unex.es; 3Grupo de Investigación EFISAL, Universidad Autónoma de Chile, Sede Talca 3467987, Chile; emerellanon@uautonoma.cl

**Keywords:** physical literacy, exercise, self-perceived fitness, physical fitness, health status, Fitness Perception Scale for Adolescents (FP VAS A)

## Abstract

**Simple Summary:**

Alarming data on physical inactivity and sedentary lifestyles in children and young people are increasing. The level of physical fitness is considered a predictor of morbidity and comorbidities resulting from physical inactivity. Physical literacy, which includes the assessment of physical fitness, could be considered a crucial element for understanding adolescents’ health status and physical activity-related behaviours. Moreover, the self-perceived physical fitness should also be considered. Thus, this study analysed the relationship between physical literacy and self-perceived physical fitness in children and adolescents aged 8–12 years. It could be concluded that a higher level of self-perceived physical fitness would be associated with a greater level of physical literacy.

**Abstract:**

Background: Physical fitness is considered a predictor of mortality and comorbidities resulting from physical inactivity in children and adolescents. Physical literacy provides a robust and comprehensive assessment of physical fitness. Thus, it is considered a crucial element for understanding children and adolescents’ health status and their physical activity-related behaviours. Moreover, the self-perceived fitness should also be considered. Therefore, this study aims to analyse the relationship between physical literacy and self-perceived physical fitness in children and adolescents aged 8–12 years. Methods: A single-measure cross-sectional study was conducted with 135 children and adolescents. Anthropometric measurements and the Canadian Assessment of Physical Literacy Development (CAPL-2) were applied. Descriptive statistics and correlations were calculated. The Fitness Perception Scale for Adolescents (FP VAS A) scale was also administered to assess participants’ self-perceived fitness. Results: Moderate significant correlations were found between “physical competence”, “motivation and confidence”, and the total CAPL-2 score with FP VAS A. Conclusions: The influence of motivation on self-perceived fitness has been demonstrated. Moreover, cardiorespiratory fitness could be considered one of the most relevant and influential factors in the total CAPL-2 score. It means that higher levels of self-perceived fitness would be associated with greater levels of physical literacy.

## 1. Introduction

Childhood obesity is considered one of the main public health problems nowadays [1]. Numerous studies have analysed the prevalence of childhood overweight and obesity in Spain: e.g., the ALADINO 2019 study [2], with a prevalence of overweight and obesity in the Spanish children population aged 6–9 years of 23.3% and 13.7%, respectively; the PASOS study [3], which shows a prevalence of 20.7% of children and adolescents aged 8 to 16 years (both inclusive) who are overweight and 14.2% who are obese; and the Spanish National Health Survey [4], which indicates that the prevalence of childhood obesity now reaches more than one in ten children (10.3% in 2–17 year olds).

Obesity may be due to several factors including physical inactivity as well as high levels of sedentary lifestyles [5]. Several studies have reported alarming data on physical inactivity in children and young people [4,6]. Data such as the fact that 35.3% of the population between 15 and 69 years of age does not reach the level of healthy physical activity recommended by the WHO should be considered [4]. It should be added that 49.7% of Spanish schoolchildren maintain a completely sedentary lifestyle or only engage in very sporadic physical activity [6]. Consequently, there is a high risk of short- and long-term health problems arising from physical inactivity [7]. In fact, both physical inactivity and sedentary behaviour in children and adolescents are beginning to be considered as two determinants of adult obesity [5].

Physical fitness in childhood and adolescence is considered a key marker of health status [8] and a predictor of mortality and comorbidities resulting from physical inactivity [9,10]. Thus, assessing physical fitness during the school years may be crucial for the prevention or early detection of pathologies associated with excess body fat. Physical fitness assessments provide information on a subject’s physical condition and the effects of physical activity practices on it [11], which might help to make decisions about the most appropriate physical activity guidelines.

There are numerous objective ways to assess physical fitness in children and adolescents, including test batteries such as the Assessing Levels of Physical Activity (ALPHA-Fitness) [12], the European Physical Fitness battery (EUROFIT) [13], or the Assessment of Health-Related Physical Fitness (COFISA) [14]. However, the subjective component based on the evaluation of self-perceived physical fitness should also be considered; it has commonly been assessed through the International Fitness Scale (IFIS) [15]. Nevertheless, a novel visual analogue scale named the Fitness Perception Scale for Adolescents (FP VAS A) was recently designed for this aim. This scale presents several advantages compared to others like IFIS, which are based on a Linkert scale. FP VAS A is easier to use [16,17], more responsive (detection of clinically significant variations), and may be more valid and reliable [17,18,19]. Furthermore, it can be applied in large groups for a short period of time, facilitating the work of researchers [20].

Considering that physical activity is considered a behaviour and fitness a condition, the main task as a society should be to promote physical activity to change that condition or fitness level [21]. In children, increasing physical activity alone is not enough, as several studies have already shown [9,22]. This is due to the fact that the risk of cardiovascular diseases in the future is conditioned by the level of physical fitness [10].

In this context, the Canadian Assessment of Physical Literacy (CAPL-2), developed in Canada, provides a robust and comprehensive assessment of physical fitness [23], being one of the most innovative and important initiatives to prevent and combat childhood obesity. Physical literacy can be described as the ability and motivation to harness motor potential, contributing significantly to quality of life [24]. Thus, the comprehension of this concept may be of particular interest in analysing why children participate or not in physical activity [25]. Furthermore, a high correlation between obesity prevalence and the level of physical activity has been shown previously [5,26]. Therefore, studying the influence of obesity on physical literacy may promote more active lives in overweight or obese children.

Previous research has reported a direct relationship between objectively-measured fitness (using CAPL-2) and self-perceived fitness [15,27]. This means that a greater self-perceived level of physical fitness would be associated with a higher level of physical literacy. It should be noted that there is a lack of scientific evidence assessing the relationship between the level of physical literacy and the self-perceived fitness. Thus, due to the importance of physical fitness assessment to know the health status of children and adolescents as well as the behaviours that influence their physical activity practice [28], this study aims to explore if there is a relationship between the level of physical literacy and self-perceived physical fitness in children and adolescents aged between 8 and 12 years old.

## 2. Materials and Methods

### 2.1. Study Design

A single-measure cross-sectional study was conducted.

### 2.2. Ethics Approval

The study was approved by the Bioethics and Biosafety Committee at the University of Extremadura, according to the guidelines of the Helsinki Declaration (references codes: 10/2020; 23/2021). Prior to starting any study procedure, all participants and their parents or legal guardians signed an informed consent form accepting their participation in the study.

### 2.3. Sample Calculation

A total of 72 participants were needed to reach 95% power to detect a difference of 0.31 between the null hypothesis correlation of 0.29 (very low or close to zero association) and the alternative hypothesis correlation of 0.60 (high association) [29]. The significance level was set at an alpha level equal to 0.05.

### 2.4. Participants

To be included in the study, participants needed to meet the following eligibility criteria: (1) age between 8 and 12 years old; (2) authorised by their parents or legal guardians; (3) agreed to participate in the study; (4) not suffer any pathologies that prevent their participation in physical fitness tests or practices; and (5) not present any condition that precludes their participation in the study.

#### Sample Size

A total of 135 children aged between 8 and 12 years (non-overweight: 83 (61.5%); overweight: 52 (38.5%)), 63 males (46.7%) and 72 females (53.3%), were included. Participants were recruited through different educational centres in Extremadura (Spain). Figure 1 shows the flow chart of participants selection.

### 2.5. Procedures and Measures

Several tools were used to assess physical literacy, body composition, and physical fitness. First, the procedures included in the WOMO study protocol [30] were followed to analyse the results derived from monitoring the overweight and obesity of children and adolescents, as well as their lifestyles. All participants had the opportunity to practice the physical tests before taking any study measurement during a familiarization session.

#### 2.5.1. Anthropometric

Test conditions were standardised based on the WHO proposal [31] and those previously followed during COSI [32] and ALADINO [33] studies. Participants were asked to remove their shoes, socks, heavy clothing (coats, jackets, sweatshirts, etc.) and accessories (earrings, bracelets, rings, etc.) before the assessment. The following parameters were considered:
Bodyweight and fat mass percentage. They were measured with a bioimpedance meter (Tanita MC-780 MA, Tanita Corporation, Tokyo, Japan). Bioelectrical impedance analysis (BIA) technology has shown an almost perfect reproducibility of body fat percentage in studies with children and adolescents, making it an applicable research tool in the analysis of changes in body composition at different times [34,35,36]. Furthermore, fat mass and fat-free mass correlate almost perfectly [35].The standard assessment was performed considering participants’ age, sex, and height. The unit of bodyweight was recorded in kilograms, with a 100 g approximation. The protocol to be followed for a proper BIA measurement was 3 h after waking up and after the last meal. In addition, the individual should not have eaten or done much exercise during the previous 12 h. It is also important to urinate before starting the assessment.Height. It was measured with a measuring rod (Tanita Tantois, Tanita Corporation, Tokyo, Japan) placed on a flat surface in an upright position and perpendicular to the ground. It was measured in centimetres to the nearest millimetre. Participants had to stand upright, with shoulders balanced and arms relaxed along the body.

In addition, the body mass index (BMI) was calculated by dividing the kilograms of weight by the square of the height in metres (BMI = weight (kg)/height (m^2^).

#### 2.5.2. Physical Literacy

Physical literacy was assessed using the Canadian Assessment of Physical Literacy Development (CAPL-2) [37] that contains four domains: “daily behaviour”, “physical competence”, “motivation and confidence”, and “knowledge and understanding”. The individual’s score is received for each domain, as well as an overall physical literacy score [38]. The domain of physical competence includes objective measures of physical fitness, motor performance, and body composition [23]. Therefore, the concept of physical fitness, defined as the set of physical attributes that relate to the ability to perform any type of physical activity [39], could be significantly related to this domain. Each of the tests described in each domain was given a certain score, up to a total of 100 points. This score, obtained by each participant in each test, was assigned on the basis of the scoring tables (adapted to gender and age) described in the evaluation guide “Manual for test administration: Canadian Assessment of Physical Literacy” [38].

In this sense, each domain comprises different assessment tests and the sum of the scores obtained in each one gives the total score for each domain.

1. Daily behaviour (D1). The total score consists of the sum of two components: total steps measured (25 points) using an activity wristband (Xiaomi mi Band 3, Xiaomi Corporation, Beijing, China) which recorded the steps for a full week, and a self-reported personal questionnaire regarding the days of the week that participants have exceeded 60 min of physical activity (5 points). The score can vary from 1 to 30 points, depending on the results obtained.

2. Physical competence (D2). The total score is composed of the sum of the performance on three different tests:Abdominal plank (10 points) [40]: Consists of holding the plank position for as long as possible.Progressive Aerobic Cardiovascular Endurance Run (PACER) (10 points) [41]: Allows the cardiorespiratory competence to be determined by means of a 20-metre running test (out and back) following an acoustic signal that determines the intensity of the test.Canadian Agility and Movement Skill Assessment (CAMSA) (10 points) [42]: Allows the participants to test their motor skills through an agility circuit which includes throwing, jumping, and moving actions.

The score ranges from 1 to 10 points and can add up to a total of 1 to 30 points.

3. Motivation and confidence (D3). This domain assesses the influence of both variables on being physically active. The final score is calculated from the sum of four elements: intrinsic motivation, competition, predilection, and adaptation. The score ranges from 1 to 7.5 points for each of them, with the total score of the domain ranging from 1 to 30 points.

4. Knowledge and understanding (D4) [43]. The total score for this domain is derived from the results obtained from answering five questions: four multiple-choice questions providing from 0 to 1 points and a fifth question consisting of a fill-in-the-blanks question to complete a paragraph story, with scoring from 1 to 6.

Therefore, the total score of CAPL-2 is made up from the sum of the score obtained in each of the four domains, obtaining a final score ranging from 0 to 100 points. Thus, participants will be classified into one of the following four levels, considering the total score obtained, their sex, and their age: “beginning”, “progressing”, “achieving”, and “excelling”. The beginning and progressing level refers to children who do not achieve an optimal level of physical literacy, the “achieving” level describes children who reach a minimum level, and the “excelling” level refers to children who demonstrate a high level of physical literacy [44].

#### 2.5.3. Fitness Perception Scale for Adolescents (FP VAS A)

FP VAS A consists of a visual analogue scale that assesses participants’ perception of their own fitness level based on five different items: general fitness status, cardiorespiratory fitness, muscular strength, speed–agility, and flexibility. Each item ranges from 0 (very poor level) to 10 (excellent level). A high correlation has been demonstrated between self-perceived fitness level and objective fitness level [27]. In fact, a higher level of correlation was shown between the objective fitness level and FP VAS A compared to correlation outcomes between objective fitness and the IFIS [20]. Thus, FP VAS A shows a good reproducibility and a moderate direct correlation with participants’ objective fitness level [20]. Therefore, it could be considered a valid and reliable tool for the assessment of self-reported fitness in adolescents.

### 2.6. Statistical Analyses

All information collected was tabulated in a database designed for this study. Statistical analyses were carried out using SPSS (Version 25, IBM SPSS, Chicago, IL, USA) software and personal data were kept anonymous.

Data are presented as the mean and standard deviation and median and interquartile range both for the total sample and stratified by sex.

Normality and homogeneity were tested using the Kolmogorov–Smirnov test and Levene’s test, respectively. Then, independent *t*-tests and the Mann–Whitney U test were applied to analyse between-sex differences for parametric (height, physical competence domain, and CAMSA) and non-parametric variables, respectively. Differences were considered significant at *p* ≤ 0.05.

Finally, between-variable relationships were analysed by applying Pearson’s (parametric variables) and Spearman’s correlation coefficients (non-parametric variables). The Bonferroni correction was applied based on the formula α*= α/n − 1 [45], where α* is the corrected value at which the null hypothesis should be rejected and n is the number of hypothesis pairs. Thus, the alpha level was set at 0.003 for multiple correlations between physical literacy level and the self-perception of physical fitness. Correlation values were interpreted following Cohen’s classification [29]: 0.30 to 0.59, “moderate”; 0.60 to 0.79, “high”; and ≥ 0.80, “excellent”.

## 3. Results

A total of 90 children and adolescents participated in this study: 42 boys (46.66%) and 48 girls (53.33%). Table 1, Table 2 and Table 3 show the descriptive statistics of the total sample and segmented by sex, as well as the between-sex comparison for main characteristics and FP VAS A and physical literacy outcomes. No significant differences were found between sexes for any of the variables considered.

Table 4 displays the relationship between PL domains and FP VAS A items. Results showed moderate significant correlations between the physical competence domain (D2), motivation and confidence domain (D3), and total PL score with FP VAS A. Regarding D2, the total score and the plank test were significantly associated with cardiorespiratory fitness (r = 0.398 to 0.411, *p* < 0.001). Moreover, the D2 total score also had a significant correlation with speed–agility (r = 0.369, *p* < 0.001).

Similarly, the total score of motivation and confidence (D3) as well as the competence (D2) subsection showed a moderate significant relationship with all FP VAS A items (r = 0.373 to 0.540, *p* < 0.001). Moreover, the intrinsic motivation subsection was moderately associated with cardiorespiratory fitness items (r = 0.374; *p* < 0.001) and flexibility (r = 0.383; *p* < 0.001). Finally, a moderate correlation was observed between the total PL score with cardiorespiratory fitness (r = 0.384; *p* < 0.001) and speed–agility (r = 0.391, *p* < 0.001) items.

## 4. Discussion

The aim of this study was to analyse the relationship between physical literacy and self-perceived physical fitness. The main findings of the present study focus on the strong influence of motivation on self-perceived fitness, as well as the close relationship between fitness level and PL domain scores and vice versa.

Regarding daily behaviour, referring to daily physical activity, no correlation between the total score of the domain or any of its subsections with the FP VAS A items was observed. Thus, there is no direct relationship between the self-perceived level of physical fitness and the estimated daily regular physical activity. In contrast, previous studies informed a direct relationship between daily physical activity habits and a high self-perceived level of cardiorespiratory fitness [46,47]. Specifically, Cantó et al. (2013) [46] found an increase in the self-assessment of motor competence with a rising level of regular physical activity. Similarly, Casimiro (2000) [46] showed that a good physical self-perception positively correlates with physical fitness and a good self-perception of health in those children who are active. Thus, self-perceived motor competence has been positively related to the level of regular physical activity [46,47]. The discrepancy with our outcomes may be due to the heterogeneity of the sample of the comparative studies, since Cantó et al. (2013) covered the school context, voluntary physical activity and leisure-time physical activity [46], and the methodology used by Casimiro (2000) [46] includes participants in Compulsory Secondary Education, also forming a much more heterogeneous sample than our study.

Concerning physical competence, the main results determined that children and adolescents with a high score level in the total score, as well as only in the abdominal plank resistance test, have a better self-perception of their level of physical fitness. Similarly, previous studies also reported a direct relationship between the cardiorespiratory fitness level in children [27], adolescents [15], and adults [48] with their self-perception of it. Moreover, a significant association has been shown to exist between the self-perceived cardiorespiratory fitness level of children and adolescents and all components of PL [49]. This finding is supported by the moderate correlation informed into the present study between the physical competence total score and the plank test with the FP VAS A cardiorespiratory fitness item. Thus, our outcomes confirm that there is a close relationship between muscular endurance assessments, such as the abdominal plank test, and the self-reported cardiorespiratory fitness item of physical fitness scales, as already indicated in previous research [15,20]. Therefore, the cardiorespiratory fitness could be considered one of the most relevant and influential factors in the PL total score. Additionally, a moderate agreement between the “physical competence” total score and FP VAS A speed–agility item was also observed. Similarly, previous studies showed that a higher self-perception of physical activity was associated with a greater performance in speed and agility tests in schoolchildren [28]. Therefore, these facts are supported by the observed association between high levels of physical competence, especially in certain skills such as endurance, with a greater self-perception of cardiorespiratory fitness level.

With respect to motivation and confidence, a higher correlation with all FP VAS A items. On the one hand, the competence subsection has a moderate correlation with each of the items of the FP VAS A scale. The PA competence subsection refers to a self-reported question about how active the child or adolescent is. This finding determines the significant association between a participant’s self-perception of how much PA he/she does with respect to all items of the self-perceived physical fitness questionnaire: general physical fitness, cardiorespiratory fitness, muscular strength, speed–agility, and flexibility. This finding could be explained by a study conducted with children and adolescents aged between 8 and 12 years, which corroborates the association between a high level of self-perceived physical fitness and a high level of self-perceived physical activity [50]. Specifically, intrinsic motivation showed correlations with the cardiorespiratory fitness and flexibility items of FP VAS A. Thus, a moderate correlation was established between adolescents’ intrinsic motivation and their self-perception of flexibility and cardiorespiratory fitness. It should be added that there are studies that report a decrease of self-perceived motivation to practice physical activity as the age increases [51]. This fact could be explained by the influence of self-concept and self-esteem, as well as the concern for physical appearance in terms of physical activity during adolescence, as some studies have stated [52,53]. A previous study [54] confirmed the relationship between more active adolescents and a higher level of physical self-concept, accentuated in scales of sporting ability and physical fitness [54], which may lead us to deduce that this fact is repeated in the case of the self-perception of physical fitness. These facts affirm the great influence of motivation on the self-perception of physical fitness.

Concerning the knowledge and understanding domain, no evidence was found between the score obtained in each of its questions and the items of FP VAS A. Thus, there is no association between children and adolescents’ knowledge of physical activity and their reported level of physical fitness. In this line, a previous study showed that the knowledge and understanding domain is the domain with the lowest relationship with respect to the other domains of physical literacy [23].

Finally, regarding the total CAPL score, the strength of the relationship established with the cardiorespiratory fitness items and speed–agility of the cardiorespiratory fitness self-perception questionnaire is moderate. This fact highlights the relevance of the self-perception of cardiorespiratory fitness on the total CAPL-2 score, as already mentioned in the analysis of the physical competence domain.

In summary, the results focus on the physical competence and motivation variables, for which it was found that adolescents and children who stated that they had a good level of physical fitness, especially regarding the cardiorespiratory fitness item, coincided with those who obtained the best scores in these PL domains and vice versa.

The practice of physical activity has a beneficial effect on the functions and structures of physical fitness: musculoskeletal, cardio-respiratory, haemato-circulatory, endocrine-metabolic, and psycho-neurological [55]. There is evidence of a relationship between schoolchildren’s perception of their physical activity habits and their participation in physical and sports activities [28]. Given the relationships found, one implication of the present study is to promote training–educational programs for children and adolescents, aimed at physical activity and physical fitness work, encouraging the acquisition of an active and healthy lifestyle. In addition, the results are also relevant to the health field, which makes it an area for future projections of this study. Public health institutions should consider promoting physical activity programmes for children with comorbidities derived from excess fat mass, which is detrimental for their age, and conditioning for their future state of health.

In short, adolescents’ perception of their self-perceived physical condition could have a direct relationship with the practice of physical activity. However, the cross-sectional nature of the self-reported method is considered one of the limitations of this study as causal relationships cannot be established; a self-perception questionnaire does not allow the objective level of physical activity of adolescents to be established. This fact may serve as a proposal for future studies, which could consider other, more objective methods for the assessment of physical activity.

On the other hand, it has been shown that body fat mass is directly related to the onset of pubertal maturation in both sexes [56]. In this sense, our study includes participants who present the age of onset of pubertal stage (12 years), so it could be considered another limitation of our study.

It is also important to note as a limitation that no data were recorded on the participants’ usual intake and eating habits.

In addition, a more complete assessment could be made by studying the relationship of physical activity with, for example, self-perceived health status, life satisfaction, health-related quality of life, education, culture, or environment, since it has been suggested that improving general physical fitness may have a favourable effect on positive perceptions of health. In short, assessing the relationship between physical activity and self-perceived health will undoubtedly have a positive impact on the present and future health and well-being of children and adolescents.

## 5. Conclusions

The present study found a relationship between physical literacy and self-perceived physical fitness in children and adolescents (aged 8–12 years). More specifically, the results focused on the influence of motivation on self-perceived fitness, which could be explained by the influence of self-concept and self-esteem, as well as a concern for physical appearance as related to physical activity during adolescence. Furthermore, based on the results obtained, it was shown that cardiorespiratory fitness could be considered one of the most relevant and influential factors in the total PL score. This means that a higher level of self-perceived fitness would be associated with a higher level of physical literacy. On the other hand, no direct relationship was found between self-perceived fitness level and estimated regular daily physical activity, probably due to the homogeneity of our sample. We also found no association between children’s and adolescents’ knowledge of physical activity and their reported level of physical fitness.

## Figures and Tables

**Figure 1 biology-10-01358-f001:**
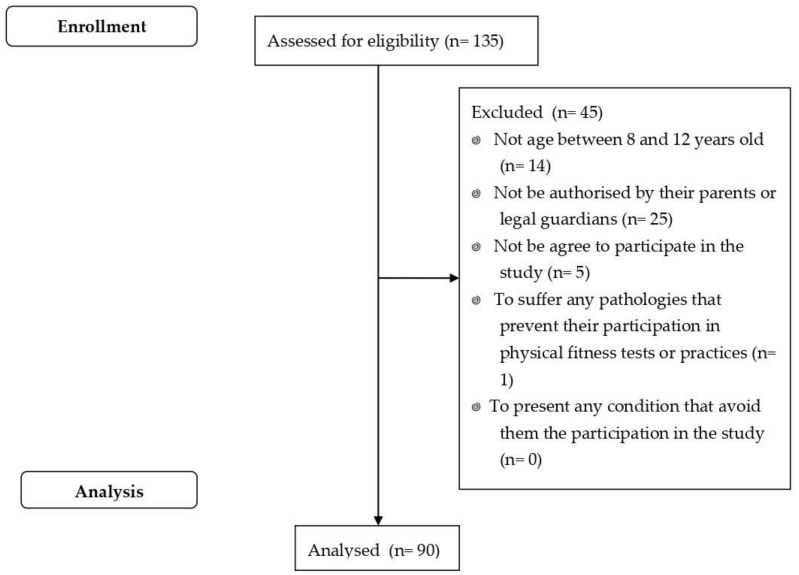
Participants selection flow chart.

**Table 1 biology-10-01358-t001:** Participants’ characteristics.

	Total	Male	Female	*p*
*n* (%)	90 (100)	42 (46.66)	48 (53.33)	
Age, years, median (IR)	11 (2)	11 (1.25)	11 (2)	0.424
Height, cm, mean (SD)	146.15 (8.3)	146.33 (7.35)	147.03 (9.12)	0.693
Weight, kg, median (IR)	40.40 (12.3)	41.1 (13.40)	39.75 (12.88)	0.416
BMI, kg/m^2^, median (IR)	18.65 (5.13)	19.75 (4.43)	18.3 (5.28)	0.320
Fat Mass, kg, median (IR)	8.95 (5.88)	9.5 (7.03)	8.85 (5.63)	0.971
Fat Mass, %, median (IR)	22.80 (9.65)	22.7 (10.08)	22.8 (9.83)	0.414

IR: Interquartile range; SD: standard deviation. The *p*-value represents the results of Student’s *t*-tests to establish differences for parametric variables and the Mann–Whitney U test for non-parametric variables.

**Table 2 biology-10-01358-t002:** FP VAS A scores for total sample and between-sex comparison.

	Total	Male	Female	*p*
*n* (%)	90 (100)	42 (46.66)	48 (53.33)	
FP VAS A-GPF, points, median (IR)	7 (4)	7 (4)	7 (4)	0.847
FP VAS A-CF, points, median (IR)	6 (3)	6 (3.25)	6 (3)	0.537
FP VAS A-MS, points, median (IR)	7 (4)	7 (5)	6 (3.75)	0.090
FP VAS A-S, points, median (IR)	8 (4)	8 (4)	7.5 (3.75)	0.350
FP VAS A-FX points, median (IR)	6 (3.25)	6 (4)	6 (3)	0.922

IR: Interquartile range; FP VAS A: Fitness Perception Visual Analogue Scale for adolescents (score 0 to 10); GPF: general physical fitness; CF: cardiorespiratory fitness; MS: muscular strength; S: speed; FX: flexibility. The *p*-value represents the results of Student’s *t*-tests to establish differences for parametric variables and the Mann–Whitney U test for non-parametric variables.

**Table 3 biology-10-01358-t003:** Physical literacy outcomes for total sample and between-sex comparisons, including their different domains.

	Total	Male	Female	*p*
*n* (%)	90 (100)	42 (46.66)	48 (53.33)	
D1, points, median (IR)	16 (13)	16.5 (13.25)	16 (11.75)	0.955
Self-reported question, points, median (IR)	3 (2)	3.71 (2)	3.5 (2.5)	0.915
Diary steps points, median (IR)	14.000 (12.000)	14.000 (12.000)	14.000 (10.500)	0.997
D2, points, mean (SD)	15.69 (6.38)	15.9 (6.58)	15.51 (6.27)	0.771
CAMSA, points, mean (SD)	5.75 (1.83)	6.02 (1.94)	5.51 (1.72)	0.186
Plank, points, median (IR)	6 (7)	6 (4.75)	6 (7)	0.733
PACER, points, median (IR)	3 (2.25)	6.25 (4)	3 (2)	0.821
D3, points, median (IR)	23.40 (2.93)	23.7 (3.65)	22.8 (2.88)	0.133
Predilection, points, median (IR)	5.6 (0)	5.6 (0)	5.6 (0)	0.470
Adequacy, points, median (IR)	5.6 (0.85)	5.6 (0.23)	5.6 (1.3)	0.151
Intrinsic motivation, points, median (IR)	6.75 (1)	6.5 (1.63)	7 (1)	0.520
Competence, points, median (IR)	6 (1.50)	6 (2)	5.75 (1.88)	0.184
D4, points, median (IR)	7 (2)	7 (2.25)	7 (2)	0.450
Question 1, points, median (IR)	1 (1)	1 (1)	1 (1)	0.355
Question 2, points, median (IR)	1 (1)	1 (1)	1 (1)	0.521
Question 3, points, median (IR)	1 (0)	1 (0)	1 (0)	0.971
Question 4, points, median (IR)	0 (1)	0 (1)	0 (1)	0.656
Question 5, points, median (IR)	4 (1)	4 (1)	5 (1)	0.557
Total PL score, points, median (IR)	72.16 (12.6)	63.47 (12.98)	62.65 (12.38)	0.760

IR: Interquartile range; SD: standard deviation; D1: daily behaviour domain; D2: physical competence domain; D3: motivation and confidence domain. D1, D2, and D3 score 1 to 30 points each; CAMSA: Canadian Agility and Movement Skill Assessment; PACER: Progressive Aerobic Cardiovascular Endurance Run; D4: knowledge and understanding domain from 1 to 10 points; total PL score: 1 to 100 points. The *p*-value represents the results of Student’s *t*-tests to establish differences for parametric variables and the Mann–Whitney U test for non-parametric variables.

**Table 4 biology-10-01358-t004:** Correlation between the domains of PL with the FP VAS A items.

		FP VAS A-GPF	FP VAS A-CF	FP VAS A-MS	FP VAS A-S	FP VAS A-FX
D1	Total	0.203	0.214	0.145	0.237	0.168
	Self-reported question	0.127	0.115	0.233	0.244	0.201
	Diary steps	0.185	0.209	0.102	0.204	0.141
D2	Total	0.315	0.398 **	0.204	0.369 **	0.138
	CAMSA	0.234	0.197	0.152	0.245	−0.109
	Plank	0.283	0.411 *	0.156	0.294	0.202
	PACER	0.217	0.255	0.131	0.304	0.103
D3	Total	0.462 **	0.473 **	0.382 **	0.462 **	0.395 **
	Predilection	0.138	0.115	−0.73	0.054	0.131
	Adequacy	0.091	0.092	0.143	0.149	0.107
	Intrinsic motivation	0.329 **	0.383 **	0.307	0.354	0.374 **
	Competence	0.497 **	0.540 **	0.471 **	0.499 **	0.373 **
D4	Total	0.071	0.131	−0.21	0.010	−0.015
	Question 1	−0.040	−0.114	−0.059	−0.083	−0.175
	Question 2	0.114	0.108	0.032	0.071	0.030
	Question 3	−0.039	−0.115	−0.136	−0.096	0.031
	Question 4	0.043	−0.036	−0.002	0.063	−0.069
	Question 5	0.062	0.312 **	0.051	0.048	0.079
Total PL score		0.342	0.391 **	0.231	0.384 **	0.197

FP VAS A: Fitness Perception Visual Analogue Scale for adolescents (score 0 to 10); GPF: general physical fitness; CF: cardiorespiratory fitness; MS: muscular strength; S: speed; FX: flexibility. D1: daily behaviour domain. D2: physical competence domain; D3: motivation and confidence domain. D1, D2, and D3 score 1 to 30 points each; CAMSA: Canadian Agility and Movement Skill Assessment. PACER: Progressive Aerobic Cardiovascular Endurance Run. D4: knowledge and understanding domain from 1 to 10 points; total PL score: 1 to 100 points. * Significant correlation for *p* < 0.05. ** Significant correlation for *p* < 0.003.

## Data Availability

The datasets used during the current study are available from the corresponding author upon reasonable request.

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
