# Peer review of "Association between Physical Literacy and Self-Perceived Fitness Level in Children and Adolescents"

_biology, 2021, doi:10.3390/biology10121358_

Round 1
Reviewer 1 Report
Thank you for the opportunity to review the manuscript “Association between physical literacy and self-perceived fitness level in children and adolescents”. The subject of the study is very important as obesity levels in children is a major public health issue.
Major comments:
The description of the performance of the tests and the procedures are not very extensive. It is not possible to understand how the different tests and domains were performed, and in what order they were performed. Could the child have a parent/legal guardian with them at the time?
Several different questionnaires and scores have been used in the study and the manuscript would have benefited from having them included in e.g. a supplementary file but they are not included in the manuscript so the reader must search to find exactly what tests that have been performed.
The issue of puberty is not discussed in this study. How many of the participants had initiated puberty? The effect of puberty on e.g. fat mass and muscular strength may influence the results.
How has participants with reading difficulties been handled in the study? Have the participants been able to have a parent/legal guarding with them to help with the reading or did a study team member help if needed?
The finding that there is no direct relationship between the self-perceived level of physical fitness and the estimated daily regular physical activity is troublesome. It needs a deeper analyse compared to the results of previous studies and not only that their samples were more heterogenous than the one included in the study. Would the results be affected by age or gender?
The sample size was a total of 135 children, but only 90 children are included in the results. What happened to the 45 children in the sample size but not included in the study? Did these children differ from the included children in anyway and why were they excluded?
Minor comments:
Rows 81-83 are unclear and needs rephrasing.
Rows 95 – 101 in the introduction are maybe better suited in the Material and Methods section?
Line 113: a full stop to end the sentence is missing.
Line 118: a full stop to end the sentence is missing.
Section 2.4: it needs to be clarified what pathologies would prevent the child’s participation in the fitness tests or practices as well as what conditions that would render them unable to participate in the study.
How did the recruitment take place? By advertisement or by word of mouth?
Section 2.5.2 please see major comment, but the different domains need to be described in more details i.e. for how many days was the activity wristband used (weekdays or weekends), for how long was the abdominal plank to be performed, what was included in the PACER and CAMSA tests and how where they scored? What are the specific questions included in domain 3 and 4?
How is the scoring of the CAPL-2 performed? What were the cut-offs used for the different levels?
Section 2.5.3 How was the individual scoring on the visual analogue scale converted into numbers?
Author Response
Dear reviewer,
You can find the authors' response in the attached file.

Reviewer 2 Report
This study aims to analyze the relationship between physical literacy and self-perceived physical fitness in children and adolescents aged 8-12 years. They revealed that higher levels of self-perceived fitness would be associated with greater levels of physical literacy. In general, the study design is fine for this study. However, the relationship between physical literacy and self-perceived physical fitness in children had been previously reported in previous studies (https://doi.org/10.1371/journal.pone.0203105, https://doi.org/10.1186/s12889-020-8318-4, DOI:10.1016/j.jesf.2018.03.002 and others). Thus, the novelty of this submission is limited. I don’t suggest accepting this submission.
- As my comment, the novelty of this submission is limited.
- No flow chat to present this study.
- No limitation was presented in this study.
- English editing is high suggested.
Author Response

(The authors gave the same response as above.)

Reviewer 3 Report
The manuscript is very well written and the structure is adequate. The methodology used is sufficient. The results are adequately displayed and support the discussion. However, I have the following comments.
I. Major Comments:
1. In the introduction I suggest including background information regarding the role of food and diet in gaining body weight.
2. In the discussion it is important to discuss i) the effect of food intake and body weight gain, and ii) interaction between diet, physical activity and body weight gain.
3. Finally, considering the results, it is necessary to discuss what the projections of this study would be, especially health interventions.
II. Minor Comments:
1. Improve the writing of the study objective.
2. Improve the legends of the tables. For example. Table 1. General characteristics of the subjects or other title.
3. Separate the results from table 1. The results on behavior and motivation should be in another table.
Author Response

(The authors gave the same response as above.)

Round 2
Reviewer 2 Report
My questions had been addressed. This submission is acceptable.
Reviewer 3 Report
Authors made all changes suggested. Therefore, manuscript can be accepted in the present form.